

# Global agricultural lands in the year 2015

Zia Mehrabi[1,2,3**], Kaitai Tong[3,4**], Julie Fortin[3,4,5], Radost Stanimirova[6], Mark Friedl[6], Navin Ramankutty[3,4]

[1]Better Planet Laboratory, University of Colorado, 4001 Discovery Drive, Boulder, CO 80303, USA
[2]Department of Environmental Studies, University of Colorado, 4001 Discovery Drive, Boulder, CO 80303, USA
[3]Institute for Resources, Environment and Sustainability, University of British Columbia, 2202 Main Mall, Vancouver, BC V6T 1Z4, Canada
[4]School of Public Policy and Global Affairs, University of British Columbia, 6476 NW Marine Drive, Vancouver, BC V6T 1Z2, Canada
[5]Department of Sustainable Use of Natural Resources, Institute of Social Sciences in Agriculture, University of Hohenheim, Schwerzstraße 40, Stuttgart 70599, Germany
[6]Department of Earth and Environment, Boston University, 725 Commonwealth Avenue, Boston, MA 02215, USA
**Contributed equally to this manuscript

*Correspondence to*: Zia Mehrabi (zia.mehrabi@colorado.edu), Navin Ramankutty (navin.ramankutty@ubc.ca)

**Abstract**

While there are many global geospatial datasets representing the extent of agriculture, they predominantly represent croplands. Only a couple of global data products represent the full global agricultural footprint, including pastures. Our own research team's most recent complete publicly available agricultural land cover dataset, including both croplands and pastures, represent

circa 2000. These data, distributed on a graticule of 5 arcminutes (~10km² at the equator), have been integrated into a considerable number and diversity of research studies, modeling, data science and media applications. Further, users of these data have been interested in them for studying a variety of issues such as land use, food security, climate change and biodiversity loss. Here we present an updated dataset on the global distribution of agricultural lands (cropland and pasture) circa 2015 (15 years on since the initial study). Past studies that have constructed such datasets have been one-off exercises

that have been infrequently repeated due to the amount of effort required. Therefore, in this work, we developed a transparent and reproducible approach to update our data product while also enabling easier reproduction of future datasets. We distribute our 2015 product at the same resolution and formats as the prior product, and accompany it with a full set of replicable code and data for reconstruction. In this article we explain how the data was constructed, with links to the permanent DOIs where the data can be readily downloaded by the user community (Mehrabi et al., 2024; DOI: 10.5281/zenodo.11540554).

**1 Introduction**

Global studies incorporating human land use in Earth systems analysis require a base data layer of the extent of agriculture on the terrestrial surface. Some global agriculture data layers have received more development effort than others. For example, a



wide range of global cropland extent products now exist; built from crowdsourcing, satellite data, data fusion of survey and satellite data, and at a wide range of resolutions, spanning 10m – 10km (Di Tommaso et al., 2023; Kim et al., 2021; Van Tricht

et al., 2023). This allows for intercomparison between methods, models, and sources of data, for scientists to estimate different sources of uncertainty in their results; and ultimately for different products to be used for different downstream applications. However, despite these advances in cropland mapping, there remains much uncertainty in global estimates of cropland area, particularly for products based on remote sensing alone (Tubiello et al., 2023). Furthermore, global data on pastures and rangelands (or grazing lands) are much less well developed, partly because pasture is such a difficult land use category to

define (e.g., see Ramankutty et al., 2008). Some datasets do however exist, including HYDE from 10 000 BCE to 2015 CE (Klein Goldewijk et al., 2017) and HILDA (Winkler et al., 2021). One product, developed using an integration of satellite and census data, and covering both cropland and pasture, was publicly released in the year 2008, and represented the land circa 2000 (Ramankutty et al., 2008); Ramankutty2008 hereafter). Ramankutty2008, has been deployed in a wide range of scientific use cases (cited nearly 2000 times according to Google Scholar), as well as widely used in the media and for science

communication, but are now two decades 'out of date'. The utility of these data are, however, that they explicitly constrain land use by different classes, and provide a full view of agricultural land use across the planet within one statistically consistent product.

The applications of Ramankutty2008 have been wide-ranging, from mapping the distribution of crops (Monfreda et al., 2008)

and the use of those for plant based versus animal product supply chains (Cassidy et al., 2013), to estimating yield gaps (Licker et al., 2010; Neumann et al., 2010) and assessing the potential for closing yield gaps (Mueller et al., 2012); identifying the impacts of climate change on agricultural production (Lobell and Gourdji, 2012); estimating sources and sinks of GHG emissions on land (Carlson et al., 2017); mapping anthropogenic biomes of the world (Ellis and Ramankutty, 2008); mapping the global human footprint (Venter et al., 2016); valuing ecosystem services (Naidoo et al., 2008), identifying biodiversity

conservation trade-offs (Mehrabi et al., 2018), economic impacts on food system policies through land use (Lee et al., 2005), and even the distribution of digital technology services and opportunities in farming (Mehrabi et al., 2021). There can be little doubt that the production of these data has been highly useful and impactful for the scientific community.

There are frequently expressed requests from the user community for updates of Ramankutty2008. One previous update was

made, but was never publicly released, although was used in some scientific publications (Samberg et al., 2016; Sloat et al., 2018). Here we publicly release an update using the most recently available agricultural censuses with global coverage – a dataset of global agricultural lands for the year 2015. In developing this product, we also greatly advanced our modeling approach that calibrates satellite data against the most recently available agricultural censuses with global coverage. We do so in formats and resolution matching the original product that allow easy integration into existing analysis pipelines, models,

and applications. One difference is we do use input data at a coarser resolution than in previous efforts, but with the benefit of much more rapid acquisition and ease of future updates by others. But a note of caution: as data and methods have changed

substantially from our earlier product (representing year 2000), and in line with recommendations from the MCD12Q1 user guide, the two products should not be compared to infer change over time.

In addition to releasing the data product, we also, for the first time, release all underlying data and code for reproduction of the data. This allows this product to be easily updated by the community, for example to match the release schedule of new agricultural censuses. While the updated pipeline supports two versions, with and without calibration to national statistics from the UN Food and Agricultural Organisation (hereafter FAOSTAT calibration), we present the FAOSTAT calibrated one in this manuscript to align with the mainstream approach followed by many researchers in their work. Below we explain how the
source data was collected, the modeling and processing pipeline, validation, and summaries of the final product as a peer-reviewed reference manual for users.

## 2 Pipeline overview

The data development and analysis pipeline we used is explained in the following sections. For a quick overview of these steps, i.e. data collection, pre-processing, input data (labels, features), model training, deployment and post-processing steps
please see Fig. 1A-B.





**Figure 1. (A) Data pre-processing and training pipeline; (B) Data evaluation and post-processing;** *GDD: Growing Degree Days; GBT: Gradient Boosting Tree*

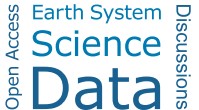

## 3 Input data

### 3.1 Agricultural inventory data

We compiled global cropland and pasture extent data from agricultural inventories and censuses over 2013-2017 (to represent circa 2015), following methods described in Ramankutty et al. (2008). Briefly, we first compiled national statistics for cropland area ("arable land" and "land under permanent crops") and pasture area ("land under permanent meadows and pastures") from UN FAOSTAT (https://www.fao.org/faostat) for the years 2013-2017, and took the mean of these to represent 2015. These

data represented a national base layer of the absolute hectarage and proportions of cropland and pasture, which we then went on to replace with subnational statistics where available as explained below.

We then added subnational statistics to countries using a strategic search: (1) starting with major agricultural countries i.e. those included in the union of the 15 countries with highest global cropland or pasture area for 2015 (total 22 countries) (2)

collecting subnational data for all EU countries from EUROSTAT (https://ec.europa.eu/eurostat) (total 29 countries), and (3) finding the union of African countries with the highest cropland or pasture area, and selecting the top 10 countries of that union (which we found to be poorly represented in steps 1-2) (total 18 countries). Our resulting list consisted of 62 unique countries covering 81.6% of global cropland and 82.1% of global pasture area.

With our priority countries in hand, we searched each of these countries' national census bureau, ministry of agriculture, statistics office or other government entity websites for agricultural censuses or statistical yearbooks circa the year 2015 (our target was 2013-2017; in 12 cases where census data was not available in that range, we used data as early as 2007 or as late as 2018).

In each census or statistical yearbook, we searched for administrative level 1 information (i.e., one level below national) on the total area of cropland and pasture. This was strategic, as it allowed for increased speed in data acquisition over prior work using exhaustive search at highest resolution census input data possible. When necessary, we translated entire documents using Google Translate's document upload feature. We included cropland areas described as "arable land", "land in crops", "fallow land", "cultivated land" and "temporary meadows". Our definition for pasture encompassed "permanent meadows", "grazing

land", "pasture land". We then extracted relevant tables and converted all units to hectares. Note that we could not find publicly available agricultural inventory data for some countries from our list during our search years, or found information on cropland area but not on pasture area; these countries were excluded from the model (Table A1). In total we found 49 countries that fit our criteria with subnational data, covering ~73% of the cropland and ~63% of the world's pasture.



## 3.2 Satellite data

We used the Moderate Resolution Imaging Spectroradiometer (MODIS) Land Cover Type (MCD12Q1) Version 6 at 500 m resolution (Sulla-Menashe and Friedl, 2018). We specifically selected the "Land Cover Type 1" layer, which labels land cover class in each pixel using the International Geosphere-Biosphere Programme (IGBP) classification scheme; see Table 3 in Sulla-Menashe and Friedl (2018) for the class definitions). We applied a temporal mode scheme to derive the most common land cover over 2013-2017, as being the representative land cover for 2015 (the mode is designed to account for interannual

fluctuations and noise in the data). A copy of the input land cover data used in the analysis is shown in Fig. 2C. The land cover map has a size of 43200 x 86400 under EPSG:4326.

## 3.3 Pre-processing

FAOSTAT serves as the national baselayer for our analysis, containing a total of 223 country level observations, which we

denote as the set $F = \{C_n\}$ where $n \in \mathbb{N}_{223}$. Each element of C in the set F represents a unique country level observation. Each country with subnational level data has multiple admin level 1 observations in a country, we denote this set as S={Dm} with K admin level 1 units, $D_m = \{s_1, ..., s_K\}$ where $m \in \mathbb{N}_{49}$ for 49 country records. Data source details are shown in Table A1. Note, that pasture definitions for Saudi Arabia are massively different between the FAOSTAT and subnational statistics, and it is therefore removed from this set (although we make predictions for it, see later), see Ramankutty2008 for a discussion

of this.

The first step in the pre-processing pipeline is to decide whether to apply a calibration to match subnational statistics to the FAOSTAT reported national values.  We optionalize in our code base different possible versions of this data in which all subnational statistics are calibrated to the FAOSTAT (i.e. where the national statistics are considered truth; as presented in the

main text) or none (i.e. where subnational data are considered the truth), so users can reproduce the data to match all, none, or a given subset of countries to the FAOSTAT totals. We distribute the all calibrated version – as this is the version which our users most frequently use. The calibration process is as follows. It is given that $D_m \in C_n$ where a country record occurs in both FAOSTAT and subnational census set, and so a factor is formulated for any outcome of interest as $C_n / \sum_{i=1}^{K} s_i$ if calibration is set true, otherwise 1. This factor will then be multiplied to each sample in set $D_m$.  After calibrating the censuses set, we merge

it with the FAOSTAT set, with the dataset formulated as $F' = \{C_n | n \notin P\} \cup \{D_m | m \in P\}$ where $P = F \cap S$.

Second we apply two spatial filters to further process the data prior to modeling: an NaN filter and Growing Degree Day (GDD; base 5ºC) (SAGE, 2022) filter. The purpose of the NaN filter is to remove any data sample that has no data (or NaN) for the cropland or pasture percentage label. Our approach involves conducting evaluations for each subnational census sample.

If the total geographical area of administrative level 1 units with missing cropland or pasture percentage label exceeds 30%





the total geographical area of the country, FAOSTAT level data will be used instead for that country and the subnational data excluded from the model. Otherwise, the available subnational census records will be utilized. These samples with partially missed labels are not usable for training, as our model relies on a complete probability distribution for each observation, which will be discussed in detail in the next section.


The GDD filter retains any sample that lies within a GDD mask (see Fig. 2). We follow similar criteria as Ramankutty2008, whereby any non-cropland in MCD12Q1 (not the mosaic classes) above 50ºN that has less than 1500ºC·d GDD is assumed to be too cold for agricultural production. Since observations (administrative units) can be partially covered by the GDD filter, we also introduce an acceptance ratio for the inclusion of an observation. For a given sample, either admin level 1 or country
level, if the ratio between the area included after the GDD filtering step (i.e. it includes some portion of the area above 1500ºC·d) and the total area of that sample which is unmasked is less than our acceptance ratio (0.95), that sample is removed.

The processed and masked dataset for cropland and pasture, containing 715 administrative units (174 admin level 0, 541 admin level 1), is shown in Fig. 2A and B respectively, where admin level 1 units removed by each filter are marked with different
color codes.

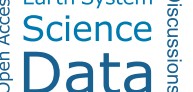



**Figure 2. (A) Subnational and FAOSTAT merged input cropland after applying NaN and GDD filters; (B) Subnational and FAOSTAT merged input pasture after applying NaN and GDD filters; (C) MCD12Q1 land cover map**



To convert the dataset into a format for modeling, we add 17 attributes representing the percentages of 17 land cover types from MCD12Q1 product for each observation. The model output labels are percentages of cropland, pasture and other land (neither cropland nor pasture) in a given observation. We also include an observation weighting column for each row using

the total geographic area of each observation. A higher weighting factor will give that corresponding observation more weight during model fitting.

Mismatches between subnational and national statistics are well known (Ramankutty et al., 2008; Ramankutty and Foley, 1998). Since we know that a substantial proportion of our user group desire consistency with FAOSTAT, we distribute data

calibrated to FAOSTAT in the main context. For cases where sum of cropland and pasture exceeds 100% due to calibration, these subnational observations were linearly scaled to probability distribution prior to training. As we have explained above we have factored our code in a way that makes it easy to update these parameters for more specific use cases.

## 4 Model

### 4.1 Set up

We modeled the relationship between the proportion of cropland, pasture and other in an administrative unit to the proportion of each satellite-based land cover in those units. We used this model to downscale the proportion of each agricultural land use onto a gridded surface taking advantage of the higher spatial resolution of the satellite data. The basic model we employed was a gradient boosting tree (GBT), with a weighted multinomial logistic loss function defined in equation (1). The GBT implementation we use adopts a one-vs-rest classification approach, where 3 models fk(x) are trained for each class label

(cropland, pasture, other).

$$L_i = \sum_{k=1}^{K} -\omega_i (y_{i,k} \log(\hat{y}_{i,k})) \qquad (1)$$

In the loss function (1), we define $i = 1,2,3,...,N$ for $N$ total number of training samples, and $\omega$ is the weight assigned to each sample based on the geographic area in that administrative unit. $y_{i,k}$ is the census-derived probability for sample $i$ in class $k$,

and $\hat{y}_{i,k}$ is the predicted probability. The overall predicted probability for that sample i can be expressed in terms of the softmax of model $f_k(x)$ in (2).

$$\hat{y}_{i,k} = \sigma(f_k(x_i)) = \frac{e^{f_k(x_i)}}{\sum_{l=1}^{K} e^{f_l(x_i)}} \qquad (2)$$

We use this model for a number of practical reasons: first is its ability to produce stable predictions despite multicollinearity

in the predictor matrix (unlike a linear model estimated using least squares); the second is its ability to capture higher order interactions amongst the predictors without need for pre-specification. Our choice of loss function was driven by the



biophysical constraint that the proportions of different land classes within an administrative unit (e.g. cropland, pastureland and other) must all fall between 0-1. We fit the model using the h2o.ai framework (h2o.ai, 2022) which is fully parallel and readily supports per-row observation weights which we use to incorporate area weighting in the model.


The five key hyperparameters (maximum tree depth=5, column sampling rate=0.5, number of trees=75, learning rate=0.1, and minimum number of observations per leaf split=5) were selected using 10-fold spatial cross-validation on the 715 administrative units. More specifically, we use a 9:1 training and testing split, where the test set is uniformly random sampled across all available geospatial units. During spatial cross-validation within the training set, each fold of the validation set is
sampled by blocks of regions that are close to one another in space. We use RMSE and $R^2$ as metrics to evaluate the initial model performance (i.e. against the test set). The results are shown in Table 1, illustrating high fits at the administrative unit level.

|  | Cropland | Pasture | Other |
|---|---|---|---|
| *RMSE* | 0.072 | 0.171 | 0.178 |
| *R²* | 0.822 | 0.349 | 0.463 |

**Table 1. RMSE and R² of trained GBT on test set**

### 4.2 Deployment

For deployment a 20 x 20 kernel is convoluted over the MCD12Q1 land cover product with stride 20 to extract 2160 x 4320 batches of block matrices. A histogram operator is then applied to each block matrix to obtain the percentage of occurrences of each land cover class in that block. Our trained model then predicts over all batches of block matrices, the proportion of cropland, pasture land and other land on a 5 arcminute (~10km x 10km at the equator) lattice. The final map has a size of 2160 x 4320 under the same EPSG:4326 projection as MCD12Q1.


### 4.3 Post-processing

For post-processing, we introduce a bias-correction step to bridge the unknown relationship between block matrix unit during deployment and administrative unit level during training. Each pixel of our output map falls within a boundary $R_n$ of a training label $y_n$ is denoted as $\hat{y}_{n_{ij}}$ for $(i,j) \in R_n$. Each $\hat{y}_n$ contains 3 channels, representing cropland, pasture and other land use
percentages. The bias-correction factor (tuple) for each pixel in $R_n$ is therefore $b_n = y_n \sum_{(i,j) \in R_n} A_{ij} / \sum_{(i,j) \in R_n} [\hat{y}_n \otimes A]_{i,j}$, where $A \in \mathbb{R}^{2160 \times 4320}$ is the global area matrix, and $\otimes$ is the element-wise multiplication symbol. This factor (tuple) is then multiplied to all pixels in $R_n$, as $\hat{y}_{n_{ij}}' = b_n \hat{y}_{n_{ij}}$. In simple terms, we use this post-processing step to ensure convergence between the pixel-level deployment and the administrative unit-level reported values for geographies where that data exist.

To maintain the probability distribution we further apply a scaling operator to each pixel to force the sum of factored proportions back to 1. The operator is formulated as $\hat{y}_{n_{ij}}' \leftarrow \hat{y}_{n_{ij}}' / \sum \hat{y}_{n_{ij}}'$.

To remove boundary artifacts between administrative units, we then apply pycnophylactic interpolation (Tobler, 1979) with relaxation at the end of each bias-correction iteration on all weights $b_n$. The property of pycnophylactic interpolation ensures
the regional sum remains unchanged after smoothing, which does not interfere with the effectiveness of bias-correction steps. Specifically, the mean filter in this process we used is [0.5, 0, 0.5], with a converge value of 3 and relaxation 0.2.

The spatial patterns of predicted outcomes within a subnational unit result from the cross-validated model, hence are built maximizing the bias-variance trade-off. We do however force convergence of these subnational predictions to match the input
data. Also, we note our model is global, unlike previous regionally parameterized models from the circa 2000 agricultural land product. We do this due to our focus on rapidly acquiring label data at administrative level 1, rather than previous attempts, which included data down to administrative level 3. Due to the global nature of this model, a number of additional corrections are made. In each iteration of bias-correction, we apply the GDD mask, water body mask, an Australian cropland and pasture mask (ABARES, 2022) and an aridity mask (Zomer et al., 2022) to the output map to remove non-agricultural regions that
otherwise would get re-introduced by bias correction back to administrative level data. A specific mask for Australia was employed, as was previously done with Ramankutty2008, due to consistently poor performance of the globally parameterized model in that region, we apply two rules: for pasture we mask everything here as 'non agricultural land', and for cropland we mask everything 'non-agricultural land' AND grazing. Our aridity mask uses a threshold of high aridity (0.05 aridity index), used in a similar vein to the GDD mask, to remove lands unsuitable for rainfed agriculture, and updated with irrigation
equipped areas at a 1% threshold (Mehta et al., 2022) to ensure that those are maintained in the final product in highly arid regions, particularly for cropland during bias correction.

## 5 Assessment

### 5.1 Assessment at the spatial scale of administrative units

Validation of the full modeling and post-processing pipeline with the input training data was completed by aggregating our final post-processed predictions at the gridded lattice to the level of the administrative unit used in training, and comparing proportional coverage estimates to survey reported cropland and pasture proportional coverage in that unit. We undertook this validation prior, during and at the end of our postprocessing steps outlined in 4.3. Scatter plots of these comparisons are shown in Fig. 3 along with summary statistics using RMSE and $R^2$. In general, we found our model to perform well for estimating

cropland and pasture in its raw form of the deploy (i.e. with no bias-correction, iterations=0) and to converge with input data

for all 3 classes after three bias-correction steps (iterations=3).

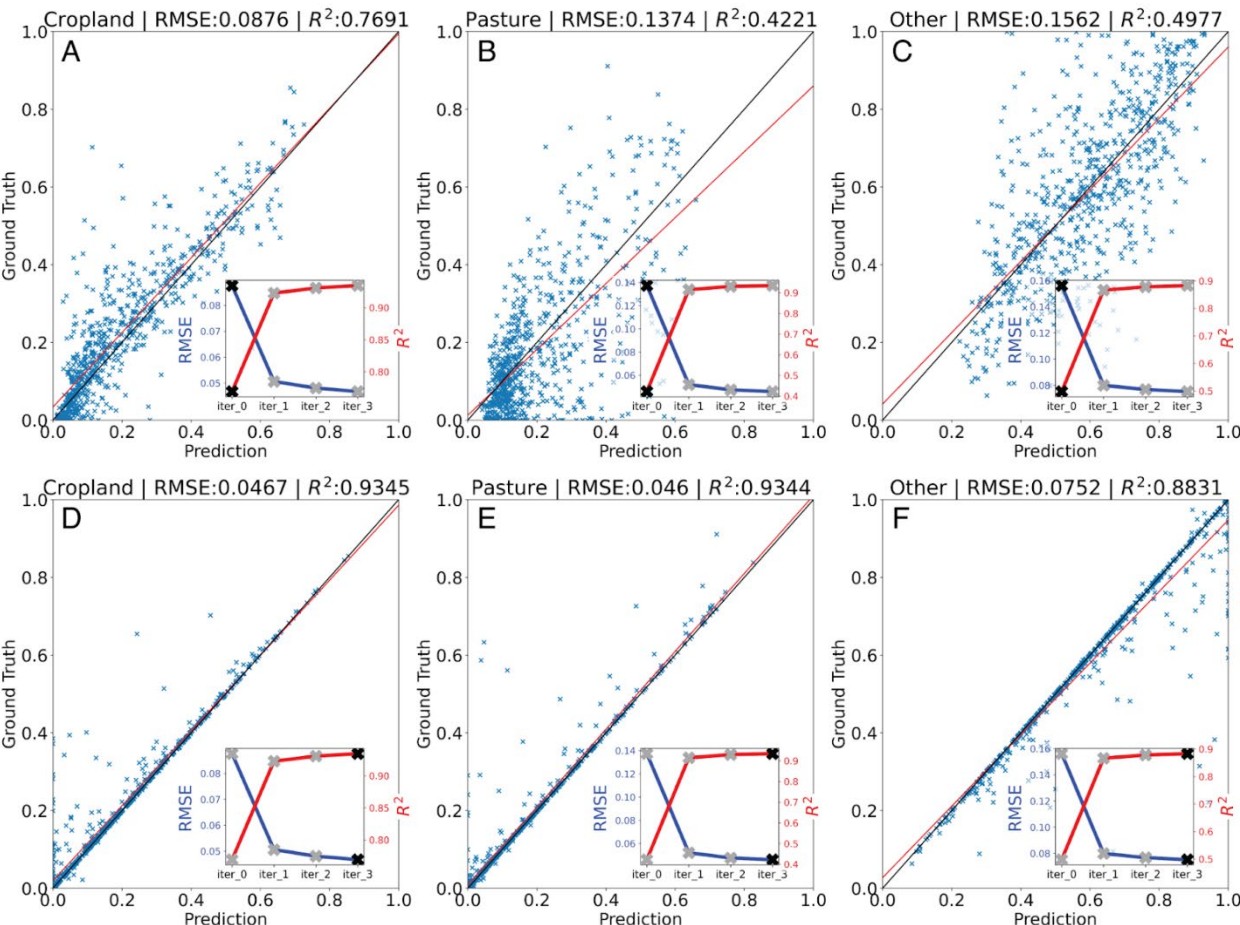

**Figure 3. Observed vs Predicted plots (scatter plot) on re-aggregated scale after bias correction iterations. Cropland, Pasture and**
**other land use; (A, B, C) Iteration 0; (D, E, F) Iteration 3**

## 5.2 Assessment at the spatial scale of predictions

We employed an independent dataset for validation of the predicted proportional land cover at the 5' level for cropland. These

data were collected through the crowd-sourced Geo-Wiki platform, in which participants identified the proportion of cropland

in nearly 36,000 sampling units of 300m x 300m, distributed around the globe (Laso Bayas et al., 2017; See, 2017). Here we

took the average percentage coverage of all Geo-Wiki observations at a given point within each 0.083 x 0.083 degree grid cell.

This validation dataset was chosen for its independence, broad geographic distribution, transparency, and critically because it



is not a modeled product itself (unlike say cropland classification products, although see below for intercomparisons with other modeled products). One thing we do note however, is that there are no global cropland products for validation or

intercomparisons at the spatial scale of the predictions that incorporate the full spectrum of croplands as we define here, GeoWiki excludes perennial crops, agroforestry plantations, palm oil, coffee, tree crops for example, and the University of Maryland product is similarly restricted to annual crops (Potapov et al., 2021).

A comparison of our predicted cropland proportional coverage and those from samples of the independent Geo-Wiki campaign

is shown in Fig. 4 by taking the difference between the common points, showing the level of agreement with our final product and this independent dataset, in terms of mean difference (0.78 percentage points) and standard deviation of the difference (27.24 percentage points). Despite the extremely close alignment on average globally, some notable differences exist geospatially, e.g. we show pixels with higher percentage cropland in the Canadian Prairies, West Africa, West India and Russia, but lower cropland in South America, South East Africa and Southern Australia. Notably no globally consistent independent

pasture data exist for external validation at the scale of predictors, although we did conduct product comparisons for both cropland and pasture to check how our predictions aligned with other independent datasets as explained below.

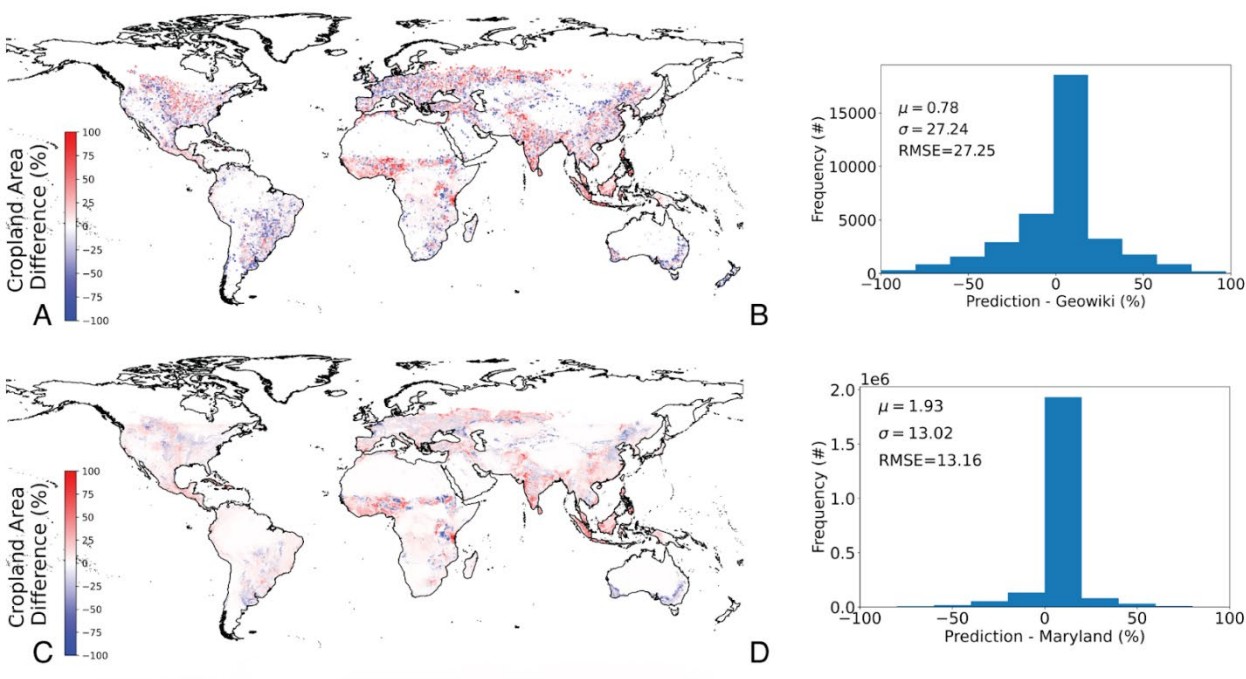

**Figure 4. Cropland external validation and intercomparisons (A) Scatter points of intercomparison against Geo-Wiki cropland data;**
**(B) Histogram of errors for Geo-Wiki comparison; (C) Map difference of intercomparison against University of Maryland cropland map; (D) Histogram of errors for University of Maryland comparison**



### 5.3 Intercomparisons at the spatial scale of gridded predictions

We conducted product intercomparisons for both our final cropland and pastureland products. For cropland we compared our
data to the University of Maryland global cropland dataset (at 30m resolution) (Potapov et al., 2021). As these data are
sequences of time ranges, we take the average coverage for 2012-2015 and 2015-2019 to arrive at a 2015 estimate of 30m
categorical cover, which we aggregated to 5' to estimate proportional coverage in each grid cell. A comparison of our 2015
estimates with the Maryland data are shown in Fig. 4, showing the agreement with the mean (1.93 percentage points) and
standard deviation of differences (13.02 percentage points). This agreement is even tighter than with the Geo-Wiki dataset.

For pastureland, we compared our predictions to two global scale pasture maps, HYDE (Klein Goldewijk et al., 2017) and
HILDA+ (Winkler et al., 2021) (Fig. 5). These products are mainly focused on land use/land cover change but also contain
static maps for the year 2015. They are both based on a satellite-based land cover map whereby classes are assigned to be
pasture, either heuristically (for HYDE), or by spatial overlap with the Gridded Livestock of the World livestock abundance
data (for HILDA+). Both are calibrated to FAOSTAT pasture statistics. We found agreement on average between our product
and these, albeit with spatial variability, with a mean difference of 5.07 (SD 25.80) percentage points with the HYDE product
and 6.00 (SD 18.74) percentage points with the HILDA+ product.

A well-known issue with pasture maps is the difficulty of defining what is a "pasture"; this could explain some of the spatial
discrepancies. For example, Fig. 5A in our global comparison with HYDE shows a large difference in Saudi Arabia, with
HYDE being calibrated to FAOSTAT values, but our model relaxing that constraint for this country. As a complement to these
global comparisons, we also examined a number of region or country specific pasture datasets in more detail, for Australia,
Brazil, the conterminous USA. These intercomparisons (Fig. A2 A-H), show the best alignment in Europe, followed by Brazil,
the USA, then Australia. These additional intercomparisons with national level datasets demonstrate broad alignment, but also
some spatial disagreement between pixel level predictions on average with those made by independent groups, models and
methods.

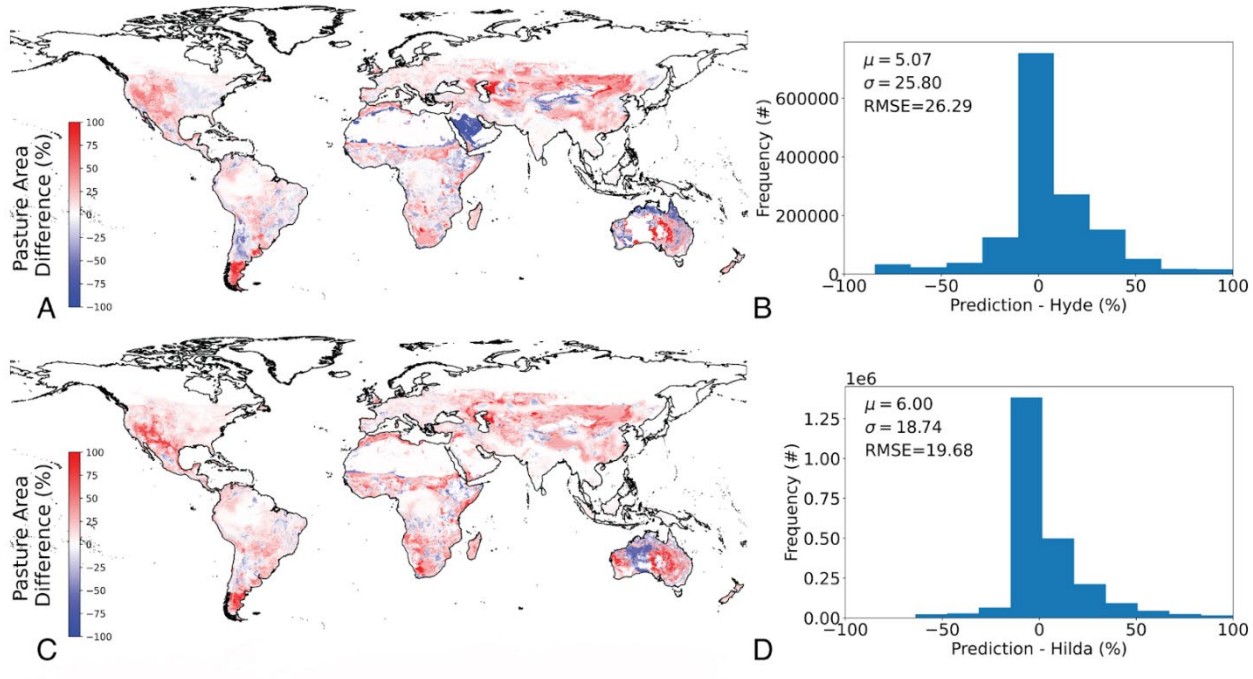

**Figure 5. Pasture map intercomparisons against (A) HYDE, (C) HILDA; Histogram of errors for (B) HYDE, (D) HILDA comparison**


## 6 Final product

Our final product of the distribution of agricultural lands for the year 2015 at 5' resolution is shown in Fig. 6A-B. For general benchmarking, we compute regional and global summaries of total cropland and pasture. Our total estimate of cropland area in the year 2015 is 1,400,700 Kha; whereas pasturelands encompass 2,774,174 Kha (compared to FAOSTAT values of

1,460,496 and 2,986,385 Kha respectively. When compared to the totals of the input data used in the model, these estimates are around 4% lower than the census dataset estimates for cropland and 7.5% lower for pasture, although geographic variation does exist for some countries and regions that deviate from these means. For example, on aggregate our product shows 8.3% lower cropland and 10.3% lower pasture in Africa than the census data totals (see Table 2 for full regional comparisons).

We note at least two sources of error a priori that likely drive these aggregate differences: (1) some residual error remains as shown in Fig. 3 after iteration 3 of the bias correction (which is assumed to also carry to locations where we don't have training data); and (2) we apply a fairly strict GDD mask for growing locations, which eliminates some administrative units where there may be agricultural lands (see Ramankutty2008 for a discussion on this), although we relax this over known satellite-classified cropland in Europe and Canada to mitigate this.


One important thing to note about these data is their intentional use. As for Ramankutty2008, these data are intended for use in global modeling studies. This statement is even more important perhaps than the ~circa 2000 product, because of the global scale of the model. There are errors that result from training a model using administrative level 0/1 data and deploying at a grid cell as outlined here. And in parameterizing a single model that is applied across the entire planet. At the same time, we

have taken reasonable care to make corrections. This update is for users that require global data that covers comprehensive cropland and pasture definitions and is numerically consistent between land use estimates.

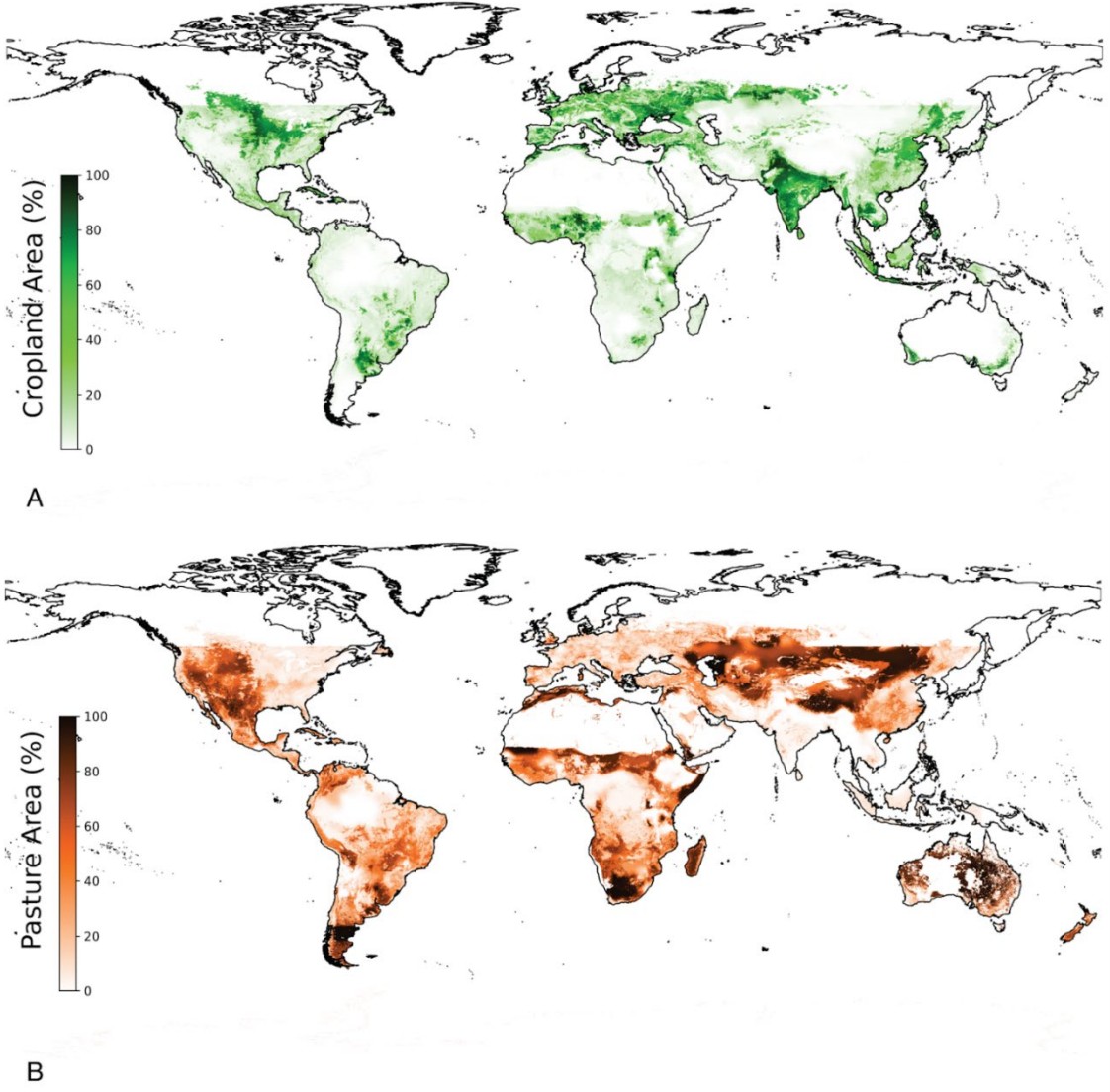

**Figure 6. (A) Cropland and (B) Pasture final product at iteration 3**



| Continent | Census (kHa) | | Prediction (kHa) - *w/ mask* | | Percentage Difference (%) | |
|---|---|---|---|---|---|---|
| | Cropland | Pasture | Cropland | Pasture | Cropland | Pasture |
| AFRICA | 271,775.92 | 821,897.46 | 250,903.10 | 745,394.96 | -8.32 | -10.26 |
| ASIA | 566,887.71 | 903,859.12 | 555,316.56 | 861,807.56 | -2.08 | -4.88 |
| EUROPE | 259,329.32 | 139,709.78 | 238,055.44 | 124,061.56 | -8.94 | -12.61 |
| L. AMERICA | 170,832.41 | 535,083.35 | 170,519.85 | 533,050.36 | -0.18 | -0.38 |
| N. AMERICA | 159,720.74 | 245,195.56 | 159,661.21 | 245,394.11 | -0.04 | 0.08 |
| OCEANIA | 31,950.60 | 340,640.00 | 26,244.08 | 264,465.90 | -21.74 | -28.8 |
| Total | 1,460,496.70 | 2,986,385.28 | 1,400,700.25 | 2,774,174.45 | -4.27 | -7.65 |

**Table 2. Summary of final product total areal estimates**

**Data availability**

The cropland and pasture data are available for download in Geotiff format at the permanent link at Zenodo (Mehrabi et al., 2024; DOI: 10.5281/zenodo.11540554), along with meta-data and instructions for use. Here you will find the FAOSTAT calibrated product (as presented in the main text) for end users, but subnational trained product could also be generated with
the provided pipeline.

**Code availability**

In addition to providing this data update, alongside this publication we also for the first time release software to enable the reproduction of this dataset as well as future updates, in a relatively easy fashion. All of the underlying training data, scripts and the trained model are stored on the Zenodo public repository link. Forks may be made from the Github repository (Add
on proofing).

We provide this material as a service to the community so that future updates, for example to the year 2020 and beyond, may be done as a community effort. Importantly, because of the streamlined pipeline, this work is easily done with modest computational resources. It takes on average 24.71 seconds for training and 2.07 hrs for deployment for each iteration and
outcome on an Apple M1 Max processor with 32 GB memory (deployment time varies significantly when changing convergence settings in pycnophylactic interpolation). This codebase resource also allows researchers to 'slot' in different land cover datasets, which may be of interest for producing finer scale predictions, e.g. with the ESA's 10m land cover dataset. While requiring higher computational capacity, this may be useful for other applications, if relevant independent test data or intercomparisons provide sufficient confidence in predictions at that scale.




## Appendix A

### Supplementary methods

We compared our pasture product to a number of independent region and country products as shown below (Figure S2). The map for Australia is the Land Use of Australia 2015-2016 at 250m resolution and was modeled based on Advanced Very High

Resolution Radiometer (AVHRR) satellite imagery and 2015-2016 census data using a Markov Chain Monte Carlo algorithm (ABARES, 2022). The map for Brazil is a 2015 land use map produced by MapBiomas at 30m resolution, using Landsat 8 satellite imagery and random forest classification (Parente et al., 2017), and was found to have an overall accuracy of 87%. The map for Europe is a 30m map of pastures for 2015, based on LUCAS (Land Use and Coverage Area frame Survey) and CLC (CORINE Land Cover) maps via a spatiotemporal ensemble machine learning (Witjes et al., 2022). The reference map

for the USA is a combination of the National Land Cover Database map for 2011 (USGS, 2011) which is based on Landsat imagery, multi-source training data and a decision tree-based classification algorithm; and the USDA rangelands map (Reeves and Mitchell, 2011), both 30m resolution. We combined these two maps for the USA because our subnational data combines data from the census (grassland pasture and range in farms) with data from the Bureau of Land Management (grassland pasture and range not in farms).




**Supplementary figures**

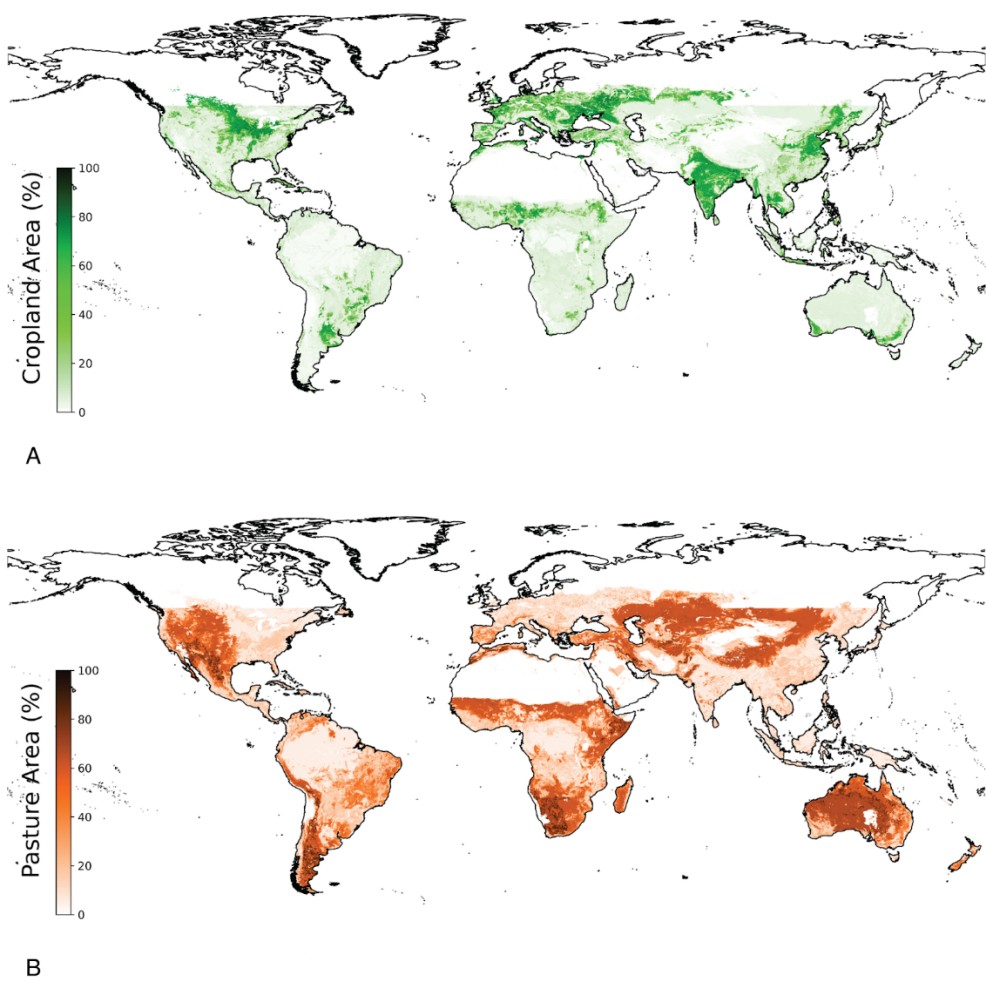

**Figure A1. (A) Cropland and (B) Pasture final product at iteration 0**






**Figure A2. Pasture map intercomparisons against (A) Brazil, (C) Australia, (E) Europe, (G) USA; Histogram of errors for (B) Brazil, (D) Australia, (F) Europe, (H) USA**



**Supplementary tables**

**Table A1. All countries for which we searched for subnational data. See main content for selection criteria for this list.**

| Country | Subnational units | Year | Institution | Source report | Source file/table | Source link | Data link | Cropland term | Pasture term | Units | Quality | Included in model training? | Notes |
|---|---|---|---|---|---|---|---|---|---|---|---|---|---|
| Algeria | - | 2010-2011 | Office Nationale des Statistiques | Recensement Economique 2011 | - | [https://www.ons.dz/spip.php?rubrique4](https://www.ons.dz/spip.php?rubrique4) | [https://www.ons.dz/IMG/pdf/agric07-11-2-4.pdf](https://www.ons.dz/IMG/pdf/agric07-11-2-4.pdf) | - | - | - | - | No | Data available, but excluded because it is not subnational. |
| Angola | - | - | National Institute of Statistics | - | - | - | - | - | - | - | - | No | Data not available. 2018-2019 census attempted but not yet completed. |
| Argentina | 23 | 2018 | Instituto Nacional de Estadistica y Censos | Censo Nacional Agropecurio 2018 | Table 3.4 | [https://cna2018.indec.gob.ar/](https://cna2018.indec.gob.ar/) | [https://cna2018.indec.gob.ar/informe-de-resultados.html](https://cna2018.indec.gob.ar/informe-de-resultados.html) | Original: superficie implantada. Translation: cropped area | Original: pastizales. Translated: pastures | hectares | Good | Yes | - |
| Australia | 7 | 2016-2017 | Australian Bureau of Statistics | Land Management and Farming in Australia 2016-2017 | File 46270DO002_201617 | [https://www.abs.gov.au/](https://www.abs.gov.au/) | [https://www.abs.gov.au/statistics/industry/agriculture/land-management-and-farming-australi](https://www.abs.gov.au/statistics/industry/agriculture/land-management-and-farming-australi) | land mainly used for crops | land mainly used for grazing | hectares | Good | Yes | - |



| Country | | Year | | | | | | | | | | | |
|---------|---|------|---|---|---|---|---|---|---|---|---|---|---|
| | | | | | | | a/latest -release | | | | | | |
| Austria | 9 | 2016 | Eurostat | Main farm land use by NUTS 2 regions | EF_L US_ MAI N | https:// ec.euro pa.eu/e urostat/ web/m ain/ho me | https:// ec.euro pa.eu/e urostat/ databro wser/vi ew/EF_ LUS_ MAIN __custo m_259 5437/d efault/t able?la ng=en | arable land + permane nt crops | permane nt grassland | hecta res | Good | Yes | - |
| Belgiu m | 11 | 2016 | Eurostat | Main farm land use by NUTS 2 regions | EF_L US_ MAI N | https:// ec.euro pa.eu/e urostat/ web/m ain/ho me | https:// ec.euro pa.eu/e urostat/ databro wser/vi ew/EF_ LUS_ MAIN __custo m_259 5437/d efault/t able?la ng=en | arable land + permane nt crops | permane nt grassland | hecta res | Good | Yes | - |
| Brazil | 27 | 2017 | Institut o Brasilei ro de Geogra phia e Estatist ica | Censo Agro 2017 | Table 6881 | https:// www.i bge.go v.br/ | https:// sidra.ib ge.gov. br/tabel a/6881 #result ado | Original: lavouras permane ntes, lavouras temporár ias. Translati on: permane nt crops, temporar y crops | Original: pastagen s naturais, pastagen s plantadas em boas condiçõe s, pastagen s plantadas em más condiçõe s. Translate | hecta res | Good | Yes | - |



| Country | | | | | | | | | | | | | |
|---|---|---|---|---|---|---|---|---|---|---|---|---|---|
| | | | | | | | | d: natural pastures, pastures planted in good condition, pastures planted in poor condition. | | | | | |
| Bulgaria | 6 | 2016 | Eurostat | Main farm land use by NUTS 2 regions | EF_LUS_MAIN | https://ec.europa.eu/eurostat/web/main/home | https://ec.europa.eu/eurostat/databrowser/view/EF_LUS_MAIN__custom_2595437/default/table?lang=en | arable land + permanent crops | permanent grassland | hectares | Good | Yes | - |
| Canada | 12 | 2016 | Statistics Canada | 2016 Census of Agriculture | Table 32-10-0406-01 | http://www.statcan.gc.ca/start-debut-eng.html | https://www150.statcan.gc.ca/t1/tbl1/en/tv.action?pid=3210040601 | land in crops excluding Christmas tree area, summer fallow land | natural land for pasture, tame or seeded pasture | hectares | Good | Yes | - |
| Chad | - | - | Institut National de la Statistique, des Etudes Econo | - | - | - | - | - | - | - | - | No | Data not available. First census beginning. |





| | | | | | | | | | | | | |
|---|---|---|---|---|---|---|---|---|---|---|---|---|
| | | | miques et Démographiques | | | | | | | | | |
| China | 31 | 2015 | National Bureau of Statistics of China | China Statistical Yearbook 2017 | Table 8-23 (cropland); Table 8-27 (pasture) | http://www.stats.gov.cn/english/ | https://www.stats.gov.cn/sj/ndsj/2016/indexeh.htm | area of cultivated land | area of grassland | kilo-hectares | Good | Yes | - |
| Croatia | 2 | 2016 | Eurostat | Main farm land use by NUTS 2 regions | EF_LUS_MAIN | https://ec.europa.eu/eurostat/web/main/home | https://ec.europa.eu/eurostat/databrowser/view/EF_LUS_MAIN__custom_2595437/default/table?lang=en | arable land + permanent crops | permanent grassland | hectares | Good | Yes | - |
| Cyprus | 1 | 2016 | Eurostat | Main farm land use by NUTS 2 regions | EF_LUS_MAIN | https://ec.europa.eu/eurostat/web/main/home | https://ec.europa.eu/eurostat/databrowser/view/EF_LUS_MAIN__custom_2595437/default/table?lang=en | arable land + permanent crops | permanent grassland | hectares | Good | Yes | - |
| Czechia | 8 | 2016 | Eurostat | Main farm land use by NUTS 2 regions | EF_LUS_MAIN | https://ec.europa.eu/eurostat/web/m | https://ec.europa.eu/eurostat/databro | arable land + permanent crops | permanent grassland | hectares | Good | Yes | - |



| | | | | | | | | | | | | | |
|---|---|---|---|---|---|---|---|---|---|---|---|---|---|
| | | | | | | | ain/home | wser/view/EF_LUS_MAIN__custom_2595437/default/table?lang=en | | | | | |
| Democratic Republic of the Congo | - | - | National Institute of Statistics | - | - | - | - | - | - | - | - | No | Data not available. Most recent census was in 1990. |
| Denmark | 5 | 2016 | Eurostat | Main farm land use by NUTS 2 regions | EF_LUS_MAIN | https://ec.europa.eu/eurostat/web/main/home | https://ec.europa.eu/eurostat/databrowser/view/EF_LUS_MAIN__custom_2595437/default/table?lang=en | arable land + permanent crops | permanent grassland | hectares | Good | Yes | - |
| Ethiopia | 10 | 2014-2015 | Central Statistical Agency | Agricultural Sample Survey 2014-2015 | Table 1 | https://www.statsethiopia.gov.et/agriculture-2/ | https://www.statsethiopia.gov.et/wp-content/uploads/2019/06/Agricultural-Sample-Survey-Land-Utilization-Meher- | all crop area, fallow land | grazing land | hectares | Good | Yes | - |





| | | | | | | | | | | | | |
|---|---|---|---|---|---|---|---|---|---|---|---|---|
| | | | | | | Season-2015.pdf | | | | | | |
| Estonia | 1 | 2016 | Eurostat | Main farm land use by NUTS 2 regions | EF_LUS_MAIN | https://ec.europa.eu/eurostat/web/main/home | https://ec.europa.eu/eurostat/databrowser/view/EF_LUS_MAIN__custom_2595437/default/table?lang=en | arable land + permanent crops | permanent grassland | hectares | Good | Yes | - |
| Finland | 5 | 2016 | Eurostat | Main farm land use by NUTS 2 regions | EF_LUS_MAIN | https://ec.europa.eu/eurostat/web/main/home | https://ec.europa.eu/eurostat/databrowser/view/EF_LUS_MAIN__custom_2595437/default/table?lang=en | arable land + permanent crops | permanent grassland | hectares | Good | Yes | - |
| France | 26 | 2016 | Eurostat | Main farm land use by NUTS 2 regions | EF_LUS_MAIN | https://ec.europa.eu/eurostat/web/main/home | https://ec.europa.eu/eurostat/databrowser/view/EF_LUS_MAIN__custom_2595437/default/table?lang=en | arable land + permanent crops | permanent grassland | hectares | Good | Yes | - |





| | | | | | | | | | | | | | |
|---|---|---|---|---|---|---|---|---|---|---|---|---|---|
| Germany | 38 | 2016 | Eurostat | Main farm land use by NUTS 2 regions | EF_LUS_MAIN | https://ec.europa.eu/eurostat/web/main/home | https://ec.europa.eu/eurostat/databrowser/view/EF_LUS_MAIN__custom_2595437/default/table?lang=en | arable land + permanent crops | permanent grassland | hectares | Good | Yes | - |
| Greece | 13 | 2016 | Eurostat | Main farm land use by NUTS 2 regions | EF_LUS_MAIN | https://ec.europa.eu/eurostat/web/main/home | https://ec.europa.eu/eurostat/databrowser/view/EF_LUS_MAIN__custom_2595437/default/table?lang=en | arable land + permanent crops | permanent grassland | hectares | Good | Yes | - |
| Hungary | 7 | 2016 | Eurostat | Main farm land use by NUTS 2 regions | EF_LUS_MAIN | https://ec.europa.eu/eurostat/web/main/home | https://ec.europa.eu/eurostat/databrowser/view/EF_LUS_MAIN__custom_2595437/default/table?lang=en | arable land + permanent crops | permanent grassland | hectares | Good | Yes | - |



| | | | | | | | | | | | | |
|---|---|---|---|---|---|---|---|---|---|---|---|---|
| India | 36 | 2019 | Department of Agriculture, Cooperation and Farmer's Welfare | At A Glance 2019 | Table 13.5 | https://agcensus.gov.in | https://eands.da.gov.in/PDF/At%20a%20Glance%202019%20Eng.pdf | land under misc. tree crops & groves not incl. in net area sown + net area sown + fallow land (total) | permanent pastures & other grazing lands | thousand hectares | Good | Yes | - |
| Indonesia | 34 | 2013 | Indonesian Central Bureau of Statistics | 2013 Agricultural Census | - | https://www.bps.go.id/ | https://st2013.bps.go.id/dev2/index.php/site/tabel?tid=66&wid=1100000000&lang=id | [sum across Planted area of rice and palawija, horticultural crops and plantations] | - | square meters | Poor | No | Data spread across multiple tables (one for each: food crops, horticulture, plantations). Does not account for fallow or multiple cropping. Crop list not comprehensive. |
| Ireland | 2 | 2016 | Eurostat | Main farm land use by NUTS 2 regions | EF_LUS_MAIN | https://ec.europa.eu/eurostat/web/main/home | https://ec.europa.eu/eurostat/databrowser/view/EF_LUS_MAIN__custom_2595437/default/table?lang=en | arable land + permanent crops | permanent grassland | hectares | Good | Yes | - |
| Italy | 21 | 2016 | Eurostat | Main farm land use by | EF_LUS_MAIN | https://ec.europa.eu/eurostat/ | https://ec.europa.eu/eurostat/ | arable land + permanent crops | permanent grassland | hectares | Good | Yes | - |





| | | | | | | | | | | | |
|---|---|---|---|---|---|---|---|---|---|---|---|
| | | | | NUTS 2 regions | | web/main/home | databrowser/view/EF_LUS_MAIN__custom_2595437/default/table?lang=en | | | | |
| Kazakhstan | 14 | 2006-2007 | Agency of the Republic of Kazakhstan on Statistics | Agriculture in Kazakhstan | - | https://stat.gov.kz/ | https://stat.gov.kz/for_users/national/agriculture2006_2007 | agricultural grounds - arable land | agricultural grounds - pastures | thousand hectares | Good | Yes | - |
| Latvia | 1 | 2016 | Eurostat | Main farm land use by NUTS 2 regions | EF_LUS_MAIN | https://ec.europa.eu/eurostat/web/main/home | https://ec.europa.eu/eurostat/databrowser/view/EF_LUS_MAIN__custom_2595437/default/table?lang=en | arable land + permanent crops | permanent grassland | hectares | Good | Yes | - |
| Lithuania | 1 | 2016 | Eurostat | Main farm land use by NUTS 2 regions | EF_LUS_MAIN | https://ec.europa.eu/eurostat/web/main/home | https://ec.europa.eu/eurostat/databrowser/view/EF_LUS_MAIN__custom_2595437/default/t | arable land + permanent crops | permanent grassland | hectares | Good | Yes | - |





| | | | | | | | | | | | | | |
|---|---|---|---|---|---|---|---|---|---|---|---|---|---|
| | | | | | | | able?lang=en | | | | | | |
| Luxembourg | 1 | 2016 | Eurostat | Main farm land use by NUTS 2 regions | EF_LUS_MAIN | https://ec.europa.eu/eurostat/web/main/home | https://ec.europa.eu/eurostat/databrowser/view/EF_LUS_MAIN__custom_2595437/default/table?lang=en | arable land + permanent crops | permanent grassland | hectares | Good | Yes | - |
| Madagascar | - | 2010 | Institut de la Statistique | Enquête Périodique auprès des Ménages 2010 | | https://www.instat.mg/ | https://www.instat.mg/documents/upload/main/MINAGRI_Annuaire_2009-2010_20-12-2012.pdf | - | - | - | - | - | No | Data not available. Only contains area of a few crops. |
| Mali | - | 2015 | Institut de la Statistique | Annuaire Statistique 2015 | - | http://www.instat-mali.org/index.php/component/content/article/11-accueil/wwwjsc-53.html | https://www.instat-mali.org/laravel-filemanager/files/shares/pub/anuair16_pub.pdf | - | - | - | - | No | Data not available. Only contains area of a few crops. |



| | | | | | | | | | | | | | |
|---|---|---|---|---|---|---|---|---|---|---|---|---|---|
| Malta | 1 | 2016 | Eurostat | Main farm land use by NUTS 2 regions | EF_LUS_MAIN | https://ec.europa.eu/eurostat/web/main/home | https://ec.europa.eu/eurostat/databrowser/view/EF_LUS_MAIN__custom_2595437/default/table?lang=en | arable land + permanent crops | permanent grassland | hectares | Good | Yes | - |
| Mauritania | - | 2015 | Office National de la Statistique | Annuaire Statistique 2015 | - | https://ons.mr/index.php/publications/statistiques | https://ansade.mr/fr/annuaire-statistiques-2015/ | - | - | - | - | No | Data not available. Only contains area of a few crops. |
| Mexico | 32 | 2007 | National Institute of Statistics and Geography | Census of Agriculture, Livestock and Forestry 2007 | File Tabulado_VIII_CAGyF_2 | http://en.www.inegi.org.mx/datos/ | http://en.www.inegi.org.mx/programas/cagf/2007/default.html#Tabular_data | Original: superficie de labor. Translation: arable land | Original: con pastos no cultivados, de agostadero o enmontada. Translation: with non-cultivated pastures (different types) | hectares | Good | Yes | - |
| Mongolia | 22 | 2015 | National Statistics Office | Report on sown area of households and enterprise, year 2015 | Table A-XAA-7 | https://www.1212.mn/ | https://www2.1212.mn/tables.aspx?tbl_id=DT_NSO_1002_003V1&SOUM_s | total sown area | - | hectares | Poor | No | Has total sown area, doesn't account for fallow |



| | | | | | | | | | | | | | |
|---|---|---|---|---|---|---|---|---|---|---|---|---|---|
| | | | | | | | elect_all=1&SOUMSingleSelect=&YearY_select_all=0&YearYSingleSelect=_2015&viewtype=table | | | | | | |
| Morocco | - | 2015-2016 | Ministère de l'Agriculture, de la Pêche Maritime, du Développement Rural et des Eaux et Forêts | Campagne Agricole 2015-2016 | - | https://www.agriculture.gov.ma/ | http://www.agriculture.gov.ma/pages/rapports-statistiques/campagne-agricole-2015-2016 | - | - | - | - | No | Data not available. Only contains area of a few crops. |
| Mozambique | 11 | 2009-2010 | Instituto Nacional de Estatistica | Censo Agro Pecuario 2009-2010 | Table 1.2 | http://www.ine.gov.mz/ | https://mozdata.ine.gov.mz/index.php/catalog/37 | Original: área cultivada. Translation: cultivated area | - | hectares | Poor | No | Glossary includes the word for pasture ("pastagen or pastagem") but does not contain a table with pasture area |
| Namibia | 14 | 2013-2014 | Namibia Statistics Agency, Ministry of Agriculture | Namibia Census of Agriculture 2013-2014 | File S3_S9_land_use_area_measurement_anonym | https://nsa.org.na/ | https://microdata.fao.org/index.php/catalog/940 | [sum across crops across households w/ hhwgt] | [sum grazing land across households w/ hhwgt] | hectares | Poor | No | Microdata: land use in variable q0302_land_use_code covers crops and grazing land. Values don't match summary in Table 3.3 of |





| | | | | | q030 2 | | | | | | | | https://d3rp 5jatom3eyn. cloudfront.n et/cms/asset s/documents /Namibia_C ensus_of_A griculture_C ommercial_ Report2.pdf . NA for grazing land for several regions. |
|---|---|---|---|---|---|---|---|---|---|---|---|---|---|
| Netherl ands | 12 | 2016 | Eurosta t | Main farm land use by NUTS 2 regions | EF_L US_ MAI N | https:// ec.euro pa.eu/e urostat/ web/m ain/ho me | https:// ec.euro pa.eu/e urostat/ databro wser/vi ew/EF_ LUS_ MAIN __custo m_259 5437/d efault/t able?la ng=en | arable land + permane nt crops | permane nt grassland | hecta res | Good | Yes | - |
| Niger | - | 2014 | Niger Nation al Institut e of Statisti cs | Annuaire Statistiqu e 2012-2016 | - | https:// www.st at-niger.o rg/ | https:// www.st at-niger.o rg/wp-content /upload s/2020/ 06/Ann uaire_S tatistiq ue_201 2-2016-2.pdf | - | - | - | - | No | Data not available. Only contains area of a few crops. |
| Nigeria | 37 | 2010-2012 | Nation al Bureau of | Agricultu ral Sector Data 2010-2012 | - | https:// www.n igerian stat.go v.ng/na | https:// nigeria. openda taforafr ica.org/ | - | - | - | - | No | Data not available. Only contains |





| | | | Statistics | | | da/index.php/catalog/52 | yktrpcf/agricultural-sector | | | | | | area of a few crops. |
|---|---|---|---|---|---|---|---|---|---|---|---|---|---|
| Pakistan | 4 | 2010 | Pakistan Bureau of Statistics | Agricultural Census 2010 | - | https://www.pbs.gov.pk/ | https://www.pbs.gov.pk/sites/default/files/agriculture/publications/agricultural_census2010/Tables%20%28Pakistan%20-%20In%20Hectares%29.pdf | farm area cultivated | - | million hectares | Poor | No | Data not available for pasture. |
| Poland | 16 | 2016 | Eurostat | Main farm land use by NUTS 2 regions | EF_LUS_MAIN | https://ec.europa.eu/eurostat/web/main/home | https://ec.europa.eu/eurostat/databrowser/view/EF_LUS_MAIN__custom_2595437/default/table?lang=en | arable land + permanent crops | permanent grassland | hectares | Good | Yes | - |
| Portugal | 7 | 2016 | Eurostat | Main farm land use by NUTS 2 regions | EF_LUS_MAIN | https://ec.europa.eu/eurostat/web/main/home | https://ec.europa.eu/eurostat/databrowser/view/EF_ | arable land + permanent crops | permanent grassland | hectares | Good | Yes | - |



| | | | | | | | | | | | | |
|---|---|---|---|---|---|---|---|---|---|---|---|---|
| | | | | | | | LUS_MAIN__custom_2595437/default/table?lang=en | | | | | |
| Romania | 8 | 2016 | Eurostat | Main farm land use by NUTS 2 regions | EF_LUS_MAIN | https://ec.europa.eu/eurostat/web/main/home | https://ec.europa.eu/eurostat/databrowser/view/EF_LUS_MAIN__custom_2595437/default/table?lang=en | arable land + permanent crops | permanent grassland | hectares | Good | Yes | - |
| Russian Federation | 83 | 2016 | Federal State Statistic Service | 2016 Russian Agricultural Census | - | https://eng.gks.ru/ | https://rosreestr.gov.ru/activity/gosudarstvennoe-upravlenie-v-sfere-ispolzovaniya-i-okhrany-zemel/gosudarstvennyy-monitoring-zemel/sostoyanie-zemel-rossii/g | Original: [subtract всего - пастбища]. Translated: [subtract Farmland total area - Farmland pasture] | Original: пастбища. Translated: pasture | thousand hectares | Good | Yes | - |



| | | | | | | | | | | | | | |
|---|---|---|---|---|---|---|---|---|---|---|---|---|---|
| | | | | | | osudarstvennyy-natsionalnyy-doklad-o-sostoyanii-i-ispolzovanii-zemel-v-rossiyskoy-federatsii/ | | | | | | | |
| Saudi Arabia | 13 | 2015 | General Authority for Statistics | Detailed Results of the Agriculture Census | File lzry_0; Table 94 | https://www.stats.gov.sa/en | https://www.stats.gov.sa/en/22 | permanent trees + date trees + open field vegetables + grain and feed + fallow + temporary meadows | permanent meadows | donum (1000 m2) | Good | No | Not included because of very large discrepancy with FAOSTAT values; see main text for justification |
| Slovenia | 2 | 2016 | Eurostat | Main farm land use by NUTS 2 regions | EF_LUS_MAIN | https://ec.europa.eu/eurostat/web/main/home | https://ec.europa.eu/eurostat/databrowser/view/EF_LUS_MAIN__custom_2595437/default/table?lang=en | arable land + permanent crops | permanent grassland | hectares | Good | Yes | - |
| Slovakia | 4 | 2016 | Eurostat | Main farm land use by | EF_LUS_MAIN | https://ec.europa.eu/eurostat/ | https://ec.europa.eu/eurostat/ | arable land + permanent crops | permanent grassland | hectares | Good | Yes | - |



| Country | | | | | | | | | | | | | |
|---|---|---|---|---|---|---|---|---|---|---|---|---|
| | | | | NUTS 2 regions | | [web/main/home](web/main/home) | [databrowser/view/EF_LUS_MAIN__custom_2595437/default/table?lang=en](databrowser/view/EF_LUS_MAIN__custom_2595437/default/table?lang=en) | | | | | |
| Somalia | - | 2014 | National Bureau of Statistics | Population Estimation Survey of Somalia | | [https://nbs.gov.so/](https://nbs.gov.so/) | [https://www.nbs.gov.so/docs/Analytical_Report_Volume_5.pdf](https://www.nbs.gov.so/docs/Analytical_Report_Volume_5.pdf) | - | - | - | - | No | Data not available. |
| South Africa | 10 | 2017 | Statistics South Africa | 2017 Census of Commercial Agriculture | Table G | [http://www.statssa.gov.za](http://www.statssa.gov.za) | [http://www.statssa.gov.za/publications/Report-11-02-01/Report-11-02-012017.pdf](http://www.statssa.gov.za/publications/Report-11-02-01/Report-11-02-012017.pdf) | arable land | grazing land | hectares | Good | Yes | - |
| Spain | 19 | 2016 | Eurostat | Main farm land use by NUTS 2 regions | EF_LUS_MAIN | [https://ec.europa.eu/eurostat/web/main/home](https://ec.europa.eu/eurostat/web/main/home) | [https://ec.europa.eu/eurostat/databrowser/view/EF_LUS_MAIN__custom_2595437/default/table?lang=en](https://ec.europa.eu/eurostat/databrowser/view/EF_LUS_MAIN__custom_2595437/default/table?lang=en) | arable land + permanent crops | permanent grassland | hectares | Good | Yes | - |



| Country | | Year | Agency | Census/Survey | Table | URL1 | URL2 | crop def | pasture def | units | quality | avail | Notes |
|---|---|---|---|---|---|---|---|---|---|---|---|---|---|
| Sudan | - | 2008 | Central Bureau of Statistics | - | - | - | - | - | - | - | - | No | Data not available. |
| Sweden | 8 | 2016 | Eurostat | Main farm land use by NUTS 2 regions | EF_LUS_MAIN | https://ec.europa.eu/eurostat/web/main/home | https://ec.europa.eu/eurostat/databrowser/view/EF_LUS_MAIN__custom_2595437/default/table?lang=en | arable land + permanent crops | permanent grassland | hectares | Good | Yes | - |
| Tanzania | 22 | 2007-2008 | National Bureau of Statistics | National Sample Census of Agriculture | Table 4.7 (smallholder); Table 3.2.1 (large scale) | https://www.nbs.go.tz/ | https://www.nbs.go.tz/statistics/topic/agriculture-census-2007-2008 | [sum across Area under Temporary/Permanent Mono/Mixed Crops + Area under Permanent/Annual Mix + Fallow] | area under pasture | hectares | Good | Yes | Note: data is disaggregated into small-scale and large scale, need to sum across both tables |
| Turkey | 81 | 2015 | Turkish Statistical Institute | Annual Statistics 2015 | - | https://data.tuik.gov.tr/ | https://biruni.tuik.gov.tr/bolgeselistatistik/anaSayfa.do?dil=en | total arable land and land under permanent crops | - | hectares | Good | No | Data not available for pasture. |
| Uganda | 14 | 2018 | Uganda Bureau of | Annual Agricultu | - | https://www.ubos.org | https://uganda.opend | total crop area | - | hectares | Good | No | Data not available for pasture. |

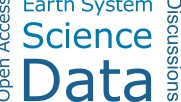

| | | | Statistics | ral Survey | | /publications/statistical/ | ataforafrica.org/zfafxee/agricultural-household-characteristics-in-uganda-at-sub-region-level-aas-2018 | | | | | | |
|---|---|---|---|---|---|---|---|---|---|---|---|---|---|
| Ukraine | 24 | 2015 | State Statistics Service of Ukraine | Agriculture of Ukraine | Table 9.22 & 9.23 | https://ukrstat.ua/en | http://www.ukrstat.gov.ua/druk/publicat/kat_e/publ4_e.htm | arable land | agricultural land - arable land | thousand hectares | Good | Yes | Cropland = Arable land is not perfect because it excludes perennial crops; Pasture = Agricultural land - Arable land is not perfect because it includes hayfields. But there is no data available at regional level that can resolve this. |
| United Kingdom | 42 | 2016 | Eurostat | Main farm land use by NUTS 2 regions | EF_LUS_MAIN | https://ec.europa.eu/eurostat/web/main/home | https://ec.europa.eu/eurostat/databrowser/view/EF_LUS_MAIN | arable land + permanent crops | permanent grassland | hectares | Good | Yes | - |




| | | | | | | | | | | | | | |
|---|---|---|---|---|---|---|---|---|---|---|---|---|---|
| | | | | | | __custom_2595437/default/table?lang=en | | | | | | | |
| United States of America | 52 | 2017 | National Agricultural Statistics Service; United States Department of Agriculture | 2017 Census of Agriculture (cropland); USDA ERS Major Land Uses 2012 (pasture) | - | https://quickstats.nass.usda.gov/results/672B19BC-9CA0-31C9-87EA-CF2003B77557 (cropland); https://www.ers.usda.gov/data-products/major-land-uses/major-land-uses/ (pasture) | https://www.nass.usda.gov/ | cropland | Grassland and other nonforested pasture and range in farms plus estimates of open or nonforested grazing lands not in farms | acres | Good | Yes | - |




## Author contributions

ZM, NR designed the study. JF, KT collected the census data. RS and MF provided the MODIS data. KT coded and implemented the pipeline, performed the analysis and model validation under supervision of ZM. JF conducted pasture map intercomparisons. KT, ZM, JF and NR discussed and interpreted results. ZM coordinated the writing of the first draft of the paper with extensive input from KT, JF, NR. All authors provided textual edits, and assisted with revisions.

## Competing interests

The authors declare that they have no conflict of interest.

## Acknowledgements

This research was supported by an NSERC Discovery Grant #RGPIN-2017-04648 and a Canada Research Chair award to NR. ZM was supported by the University of Colorado Boulder for a portion of the study. Contributions from MF and RS were partially supported by NASA grant #80NSSC18K0994.

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
