# Peer review of "Global agricultural lands in the year 2015"

_Earth System Science Data, 2024_

## Author Comment (AC2)

Link to Manuscript: https://essd.copernicus.org/preprints/essd-2024-279/essd-2024-279.pdf

ESSD Link: https://essd.copernicus.org/preprints/essd-2024-279/#discussion

RC1 Comments

**General comments:**

This manuscript presents an important update to a widely used global dataset of agricultural lands (including both cropland and pasture) for circa 2015. As a researcher working in this space, I can confidently say that these data are needed and will be widely used. As an anecdote, I have interacted with many researchers in the academic, NGO, and private space that still use the original data (circa 2000) in their analyses, as no other datasets exist that are global, comprehensive in both crop and pasture percent area, and are served at a resolution readily useable for global models and other land use and ghg accounting metrics.

This manuscript is being submitted in 2024 for maps that reflect 2015 and by the time of publication may be 10 years out of date. While it would be great to have something more updated, this delay is likely a reflection of the time it takes to receive and process census and survey data, create and document reproducible code, make comparisons with other global maps, and the general nature of academic data production. While other higher-resolution and more updated datasets have come out and will continue to come out, these 2015 data will remain very useful for the reasons stated above. Furthermore, future updates could be faster as the authors seem to have taken great care to create a reproducible pipeline for updating future versions of the maps, which is a great service to the community and will ensure reproducibility, trust, uptake, and longevity.

The methods used in the map production are tested and sound, as far as I'm aware. The bias-correction steps and post-processing methods, including pycnophylactic interpolation, seem appropriate. Because the data production pipeline is open, others can assess the impact that these steps have on the final product. The authors have made the proper statements about the appropriate use of the data ("…these data are intended for use in global modeling studies… This update is for users that require global data that covers comprehensive cropland and pasture definitions and is numerically consistent between land use estimates"). There are possibly some idiosyncrasies and missing areas in the maps, which I mention below. I make one suggestion to either adjust the latitude/GDD mask for pastures to better reflect reality or at least to document the extent of those missing areas when comparing to HYDE and HILDA+.

I am not an expert in the validation of geospatial data products so I would defer to other reviewers on this topic. I understand that the modern best practice involves an independent visual inspection, but it also seems to me that it would be extremely challenging (or not possible?) to do that on a percent area product at this resolution.

Overall, the manuscript and accompanying maps and code represent a valuable contribution to the field of global agricultural mapping.

We sincerely thank Reviewer 1 for your thorough and insightful feedback. We greatly appreciate your recognition of the dataset's significance, as well as your thoughtful suggestions for improvement. Below, we address the general comments as a whole and provide detailed responses to each specific comment and technical correction.

**General Comments**

*"There are possibly some idiosyncrasies and missing areas in the maps, which I mention below. I make one suggestion to either adjust the latitude/GDD mask for pastures to better reflect reality or at least to document the extent of those missing areas when comparing to HYDE and HILDA+."*

**Response:**

Thank you for your suggestion. In response, we have documented the total pasture area excluded by GDD masks in comparison to HYDE and HILDA on line 382.

"Our GDD masks in comparison to HYDE and HILDA do impact on differences. In total, our GDD masks remove 1,106,005 km$^2$ of area considered in Hyde (~2% of total GDD mask area, 3.5% of pasture area) and 163,865 km$^2$ in of areas considered in HILDA+ (~0.3% of total GDD mask area, 0.05% of pasture area)."

**Specific Comments**

**Comment #1**

*"I'm glad that the data production pipeline supports the production of maps that are not aligned to FAOStat, given some of the known inaccuracies. Will those maps receive a separate peer-reviewed publication? If not, I would encourage the authors to consider presenting them here in the supplemental material. I believe these maps would be valuable but far less used if not peer-reviewed. It doesn't seem like it would have to add much length to the manuscript text if the production is just a branch of the current pipeline. I should clarify that this comment is more of a personal recommendation that I think would strengthen the paper and make the non-FAO matched product more useable. I don't think it should be taken as a pre-condition for publication as the current version does stand on its own."*

**Response:**

Thanks for your suggestion. We have included the completely uncorrected (iteration 0) maps in the Supplement (Figure A1 A-B) and also in the Zenodo repository for peer review. However it is important to recognize that how maps are corrected (whether to subnational or FAO statistics) remains a challenging question that is reliant on geographic expertise. As such, we have now also updated our description of this in the introduction Line 74:

"While the updated pipeline supports options for the user to calibrate any individual country (or not) to national statistics from the UN Food and Agricultural Organisation (hereafter FAOSTAT calibration), we present the FAOSTAT calibrated one in this manuscript to align with the mainstream approach followed by many researchers in their work (although this could be relaxed if geographic expertise exists to make alternative judgements)."

**Comment #2**

*"It would be useful to define what is a pasture or cropland area for this map as well as some heuristics about what we should expect to find in these areas. Are you adopting the FAO definitions of arable land and permanent meadows and pastures and calling it cropland and pasture? Is there a defined allowable fallow period to be considered cropland? Having an explicit definition will make it easier for others to understand if this map is suitable for their use case and also help to "map" the differences between this product and others. If the definition is in the old paper I think it's worth repeating."*

**Response:**

In Section 3.1 Line 120-191, we now clarify the definitions of cropland and pasture used throughout the paper are based on FAOSTAT definitions, as in Ramankutty20008, and include these definitions in the paper and identify where there are deviations from them in subnational statistics:

"We compiled global cropland and pasture extent data from agricultural inventories and censuses over 2013-2017 (to represent circa 2015), following methods described in Ramankutty et al. (2008). Briefly, we first compiled national statistics for cropland area and pasture area from UN FAOSTAT (https://www.fao.org/faostat) for the years 2013-2017, and took the mean of these to represent 2015. These data represented a national base layer of the absolute hectarage and proportions of cropland and pasture, which we then went on to replace with subnational statistics where available as explained below.

The baseline definitions, from the FAO, are as follows:

**Cropland:** Land used for cultivation of crops. The total of areas under "Arable land" and "Permanent crops", each of which is detailed below for completeness:

- *Arable Land*. Land used for cultivation of crops in rotation with fallow, meadows and pastures within cycles of up to five years. The total of areas under "Temporary crops," "Temporary meadows and pastures," and "Temporary fallow." Arable land does not include land that is potentially cultivable but is not cultivated.
- **Temporary crops**. Land used for crops with a less-than-one-year growing cycle, which must be newly sown or planted for further production after the harvest. Some crops that remain in the field for more than one year may also be considered as temporary crops e.g., asparagus, strawberries, pineapples, bananas and sugar cane. Multiple-cropped areas are counted only once.
- **Temporary meadows and pastures**. Land temporarily cultivated with herbaceous forage crops for mowing or pasture, as part of crop rotation periods of less than five years.
- **Temporary fallow**. Land that is not seeded for one or more growing seasons. The maximum idle period is usually less than five years. This land may be in the form sown for the exclusive production of green manure. Land remaining fallow for too long may acquire characteristics requiring it to be reclassified, as for instance "Permanent meadows and pastures" if used for grazing or haying.
- **Permanent crops**. Land cultivated with long-term crops which do not have to be replanted for several years (such as cocoa and coffee), land under trees and shrubs producing flowers (such as roses and jasmine), and nurseries (except those for forest trees, which should be classified under "Forestry"). Permanent meadows and pastures are excluded from Permanent crops.

**Pasture:** Land in Permanent meadows and pastures. Land used permanently (five years or more) to grow herbaceous forage crops through cultivation or naturally (wild prairie or grazing land). Permanent meadows and pastures on which trees and shrubs are grown should be recorded under this heading only if the growing of forage crops is the most important use of the area. Measures may be taken to keep or increase productivity of the land (i.e., use of fertilizers, mowing or systematic grazing by domestic animals.) This class includes:

- Grazing in wooded areas (agroforestry areas, for example);
- Grazing in shrubby zones (heath, maquis, garigue);
- Grassland in the plain or low mountain areas used for grazing: land crossed during transhumance where the animals spend a part of the year (approximately 100 days) without returning to the holding in the evening: mountain and subalpine meadows and similar; and steppes and dry meadows used for pasture.

We then added subnational statistics for countries using a strategic search: (1) starting with major agricultural countries i.e. those included in the union of the 15 countries with highest global cropland or pasture area for 2015 (total 22 countries) (2) collecting subnational data for all EU countries from EUROSTAT (https://ec.europa.eu/eurostat) (total 29 countries), and (3) finding the union of African countries with the highest cropland or pasture area, and selecting the top 10 countries of that union (which we found to be poorly represented in steps 1-2) (total 18 countries). Our resulting list consisted of 62 unique countries covering 81.6% of global cropland and 82.1% of global pasture area. With our priority search countries in hand, we searched each of these countries' national census bureau, ministry of agriculture, statistics office or other government entity websites for agricultural censuses or statistical yearbooks circa the year 2015 (our target was 2013-2017; in 12 cases where census data was not available in that range, we used data as early as 2007 or as late as 2018).

In each census or statistical yearbook, we searched for administrative level 1 information (i.e., one level below national) on the total area of cropland and pasture. This choice of administrative level was also strategic, as it allowed for increased speed in data acquisition over prior work (e.g. Ramankutty2008) that used exhaustive search at highest resolution census input data possible. When necessary (i.e. outside the research team's language ability), we translated entire documents using Google Translate's document upload feature. We searched in these documents, for statistics that aligned with the FAO definitions above. We note reported definitions from state records are not always consistent

with the FAO, and in these cases we undertook case-case judgements on which statistics to include; all exact wordings from the source data used in the subnational statistics is included in Table A1 for full reproducibility. We then extracted relevant tables and converted all units to hectares. Note that we could not find publicly available agricultural inventory data for some countries from our list during our search years, or found information on cropland area but not on pasture area; these countries were excluded from the model (Table A1). In total we found 49 countries that fit our criteria with subnational data, covering ~73% of the cropland and ~63% of the world's pasture."

**Comment #3**

*"The Northern latitude or GDD mask may be too strict (I think more for pastures than croplands) as it seems to be masking out some areas that should be considered agriculture such as the UK and Northern Ireland, Fennoscandia, and Iceland. You could compare to results here (https://link.springer.com/article/10.1007/s10980-024-01810-6) or here (https://link.springer.com/article/10.1038/s41598-022-20095-w?fromPaywallRec=false).*

*Additionally, in Figure 5 it looks like the latitude/GDD mask was applied to HYDE and HILDA as well before making the comparison but it would be useful to see how much pasture area was included in those datasets that's excluded from this one. I would recommend either reconsidering these constraints or at least quantifying the impact by showing how much area is left out compared to other products when this constraint is applied."*

**Response:**

As noted above, we have documented the total pasture area excluded by GDD masks in comparison to HYDE and HILDA on line 376.

"Our GDD masks in comparison to HYDE and HILDA do impact on differences. In total, our GDD masks remove 1,106,005 km$^2$ of area considered in Hyde (~2% of total GDD mask area, 3.5% of pasture area) and 163,865 km$^2$ in of areas considered in HILDA+ (~0.3% of total GDD mask area, 0.05% of pasture area)."

**Comment #4**

*"Starting on line 240 it's confusing to understand how you're treating Australia and why. For example, you mention masking grazing in the cropland maps but how are the Australian grazing areas defined? Maybe it's the Abares reference but you only mention pasture and not grazing as a term when you describe it. It would be nice to give some additional details and justifications for why an aridity mask is only used in Australia.*

*Perhaps a summary of the points made in the original paper would help this section flow better and stand alone.”*

**Response:**

Thanks for highlighting the confusion here, this section was previously written poorly and we have rewritten this section to make it clearer, Lines 310-320, and included the ABARES map in the supplement in Figure A3. Notably these grazing category does include both natural and modified grazing.

"Due to the global nature of this model, a number of additional corrections are made. In each iteration of bias-correction, we apply the GDD mask, water body mask, and an aridity mask (Zomer et al., 2022) to the output map to remove non-agricultural regions that otherwise would get re-introduced by bias correction back to administrative level data. Our aridity mask uses a threshold of high aridity (0.05 aridity index), used in a similar vein to the GDD mask, to remove lands unsuitable for rainfed agriculture, and is updated with irrigation equipped areas at a 1% threshold (Mehta et al., 2022) to ensure that those are maintained in the final product in highly arid regions, during bias correction, particularly important for irrigated cropland in dry areas.

A specific mask for Australia was employed, as was previously done with Ramankutty 2008, due to consistently poor performance of the globally parameterized model in that region. For this country mask we rely on locally available land use data developed by Australian Department Agriculture Water and the Environment: the Land Use based on Agricultural Commodities at 250m 2015-2016 (ABARES, 2022) applying two simple rules: for pasture area predictions we mask everything identified by ABARES as 'Non agricultural land', and for cropland we mask everything 'Non-agricultural land' AND "Grazing". Here Grazing (see Figure A3) includes modified and natural grazing, and was introduced to primarily exclude the large extensive grazing systems in the region. Nearest neighbour resampling of the original 250m labels to 0.083 degrees prior to masking maintained broad scale ABARES cropland patterns (see Figure 6a)."

**Comment #5**

*"Line 270. You could consider adding some additional data on top of Geo-wiki or Potopov to cover some of the missing perennial crops. See here for potential data sources (https://www.wri.org/research/spatial-database-planted-trees-sdpt-version-2).”*

**Response:**

Thanks for this suggestion. We have added a sentence to the manuscript to highlight this point an cited this dataset for readers on Lines 345-352.

"Notably, newer datasets have been developed to fill this gap (i.e. which map tree crop area rather than annual crops estimates used in other cropland definitions), such as the World Resource Institutes Spatial Database on Planted Trees (SDPT) (Richter et al., 2024). On visual inspection of these additional data (not shown) we do find a spatial correspondence that indicates differences between our cropland product (which incorporates all crop types, including

trees) and other global cropland maps (such as the Maryland or GeoWiki cropland maps, which are only focussed on annual crops), can be explained by areas mapped in SDPT, particularly Indonesia, some regions of West Africa, southern Spain."

**Technical Correction**

**Comment #1**

*"Line 210 – I was unfamiliar with the term "stride" but if it's commonly understood there is no need to change it."*

**Response:**

Thank you for pointing that out. The term "stride" is commonly used and refers to the step size between kernel operations. We have added the brief definition in the manuscript. Add to Line 276.

"or deployment a 20 x 20 kernel is convoluted over the MCD12Q1 land cover product with stride (step size) 20 to extract 2160 x 4320 batches of block matrices."

**Comment #2**

*"Figure 3. I think the axes should have units of percent"*

**Response:**

Thank you for catching this. We have updated Figure 3.

**Comment #3**

*"271 – Geo-Wiki is hyphenated"*

**Response:**

Thank you for pointing this out. We have updated the manuscript.

**Comment #4**

*"290 – I think there must be a typo here and it's actually 2012-2015 and 2016-2019."*

**Response:**

Thank you for catching this. We have updated the manuscript.

**Comment #5**

*"340 – Table 2 caption should read "… area estimates"""*

**Response:**

Thank you, we have updated the manuscript.

RC2 Comments

The authors present an updated version of a global layer of cropland and pastureland for the reference year 2015. They are correct in pointing out that there are few datasets providing information about agricultural uses, especially pastures, at global or continental scales. This explains the widespread use of the previous dataset produced by the authors and fully justifies the relevance of the data presented in this work.

With that said, I believe this contribution is highly relevant and useful. I am not a technical expert and have no prior experience with most of the methods employed by the authors. Therefore, I cannot provide meaningful comments regarding the specific methodology used to produce the dataset and, consequently, its quality. I would recommend that the editor consult expert reviewers on the technical aspects of the paper to gain further insight on this point.

Response to RC2

We sincerely thank Reviewer 2 for your thoughtful and constructive feedback. We greatly appreciate your recognition of the dataset's relevance and the valuable contribution it makes to the field of agricultural mapping. We also appreciate your suggestions regarding the readability and clarity of the manuscript. We address your specific comments below.

**General comments #1**

Regarding the dataset's usability, I found it very helpful, and I appreciate the authors' comments about the specific purposes for which the dataset should be used, as well as its limitations. I would recommend that the authors elaborate on these points in greater depth, providing a dedicated section in the paper to outline the dataset's limitations, uncertainties, and the extent and contexts in which it should be used. In this regard, I would reiterate the authors' warning at the beginning of the paper about the potential temptation to compare this dataset with the one previously produced for the reference year 2000

**Response:**

Thanks for this suggestion. We use the 'Final product' section to outline these usage notes, although key uncertainties (model level, in comparison with ground truth labels, and intercomparisons with other products) are presented in Figures 4-5 and Table 6 as results. As a guidance to users we have also included additional detail in the section "Final Product" to elaborate on our previous notes, in the following Lines 403-425. We have also reiterated recommendations around time series analysis, see below:

"When compared to the totals of the input data used in the model, these estimates are around 4% lower than the census dataset estimates for cropland and 7.5% lower for pasture, although geographic variation does exist for some countries and regions that deviate from these means. For example, on aggregate our product shows 8.3% lower cropland and 10.3% lower pasture in Africa than the census data totals (see Table 2 for full regional comparisons).

We note at least two sources of error a priori that likely drive these aggregate differences: (1) some residual error remains as shown in Fig. 4 after iteration 3 of the bias correction (which is assumed to also carry to locations where we don't have training data); and (2) we apply a fairly strict GDD mask for growing locations, which eliminates some administrative units where there may be agricultural lands (see Ramankutty2008 for a discussion on this), although we relax this over known satellite-classified cropland in Europe and Canada to mitigate this.

One important thing to note about these data is their intentional use. As for Ramankutty2008, these data are intended for use in global modelling studies. This statement is even more important perhaps than the ~circa 2000 product, because of the global scale of the model, coarser input labels. There are errors that result from training a model using administrative level 0/1 data and deploying at a grid cell as outlined here. And in parameterizing a single model that is applied across the entire planet. As such we recommend regional focussed analyses to seek more fine-tuned national or regional data. Furthermore, we stress these data should not be used for time series analysis with the 2000 product due to errors in the underlying MODIS data and different modelling pipeline. At the same time, all said, we have taken reasonable care to make corrections. This update is for users that require global data that covers comprehensive cropland and pasture definitions and is numerically consistent between land use estimates"

**General comments #2**

Some information on what cropland and pastureland means in the paper and how this definition fits in the different parts of the world would be also appreciated.

**Response:**

Reviewer #1 also requested this addition, we have updated the manuscript to address this point, please see the response above. Briefly, in Section 3.1, we now clarify the definitions of cropland and pasture used throughout the paper are based on FAOSTAT definitions, as in Ramankutty20008, and include these definitions in the paper and identify where there are deviations from them in subnational statistics.

**General comments #3**

Finally, I also suggest that the authors improve the paper's readability and structure. I believe the paper would benefit from a few changes that could better highlight the authors' work. For example, the methods section could be explained in a more detailed, step-by-step manner, making it easier for users to replicate the workflow followed by the authors.

In addition, the paper needs of some language revision to avoid small mistakes (e.g. page 2, line 65)

**Response:**

Thanks for the suggestion, we have addressed these structural changes to the paper below in our responses to your specific comment on clearer presentation of the methods. We have also done additional checks to try and catch language mistakes.

**Comment #1**

*"Page 3, Section 2*

*It would be beneficial to clearly outline the different steps of the methodology at the beginning of Section 2 (pipeline overview), providing a brief explanation of each step and then directing the reader to the sections where each part of the methodology is explained in more detail.*

*Additionally, for clarity, I recommend that the authors include subsections within Section 2 for each part of the methodological process."*

**Response:**

Thank you for your suggestion. We have now updated this section to outlined the key steps taken

"In this section, we provide a high-level overview of the proposed data pipeline, which is divided into two main parts: data pre-processing and model training (Section 2.1, Figure 1) and deployment and post-processing (Section 2.2, Figure 2). Each step in the pipeline is explained below. More detailed information, including the technical aspects of the implementation, can be found in Sections 3 and 4.

**2.1 Data pre-processing and training pipeline**

The first part of the data pipeline focuses on preparing input data and training gradient boosting tree models (Figure 1). The main steps involve:

1. Data harmonization: The raw input data comes from various sources and different formats (Table A1). This step unifies input data into a standardized structure for processing.
2. Subnational census data integration: This step replaces country-level data from FAOSTAT with more granular subnational census data, where available, to enhance spatial resolution and accuracy.
3. Computing a GDD mask: A Growing Degree Days (GDD) map is generated to identify and mask regions that are unsuitable for agricultural production due to low temperatures.
4. Applying GDD mask and NaN filters to remove non-agricultural and invalid data.
5. Extract land cover percentage for each subnational unit: Land coverage is extracted as features to be used as model inputs.
6. Train GBT: A Gradient Boosting Tree (GBT) is built for training.

**2.2 Deployment and post-processing pipeline**
The second phase of analysis involves model deployment and post-processing (Figure 2), which includes the following key steps:

1. Computing land cover percentage for each 0.083 x 0.083 grid cell: the global land coverage map is segmented into grids, which are then used as inputs for the trained model.
2. Cropland, Pasture and Other area prediction: the GBT model predicts a probability distribution for each land class over each deployment grid cell.
3. Apply masks to exclude non-agricultural regions (e.g. high aridity, low GDD).
4. Compute weight matrices to match model inputs: weight matrices are computed between masked outputs and model inputs with pycnophylactic interpolation.
5. Calibrate: The smoothed weight matrices are applied back to the model predictions, refining the outputs in each iteration to calibrate."

**Comment #2**

*"Page 4*

*Separating Figure 1A and Figure 1B into two distinct figures, each with its own caption, would enhance clarity in this part of the paper."*

**Response:**

Thank you for your suggestion. Based on the feedback, we split Figure 1A and 1B, into Figure 1 and 2.

**Comment #3**

*"Page 5, Section 3.1*

*Starting from line 100, the authors should work on a clearer presentation of the information. You searched for subnational statistics across various countries, obtaining different results (no data, data but not the required data, available data). I recommend clearly explaining which countries fall into each category, and referring readers to Table A1 for those countries where useful information was found. Including all countries in Table A1 seems unnecessary, as it adds noise to the information presented in the table."*

**Response:**

Thank you for your suggestion. In our revision we have completely updated Section 3.1. With regards to clearly presenting the countries included in the analysis and those for which data

was available or not, we have intentionally chosen to include all countries for which subnational data was searched in Table A1, along with selection criteria for a full view of the process via that Table A1 to complement the text description. This serves as a reference for transparency, showing which countries were considered in the data search process and why. For every country that was not included in the modelling, a specific note is given as to why in Table A1. We hope this covers the two concerns outlined in this comment.

**Comment #4**

*"Page 5, Line 102*

*The second part of the text in brackets should be placed outside the brackets, as it provides important clarifying information."*

**Response:**

Thank you for catching this. We have updated the manuscript.

**Comment #5**

*"Page 6, Line 120*

*The last line of the paragraph can be removed, as it does not add meaningful information."*

**Response:**

Thank you, we agree. We have updated the manuscript as suggested.

**Comment #6**

*"Page 6, Line 128*

*This appreciation for the South Arabian datasets is relevant and should be moved to Section 3.1."*

**Response:**

Thank you, we agree. We have updated the manuscript.

**Comment #7**

*"Page 6, Section 3.3*

*I recommend that the authors move all relevant pre-processing steps related to input data to the sections where these datasets are introduced and explained. You may not need to explain every detail of the pre-processing workflow in the main text, but simply refer to the main steps and provide a detailed description of all pre-processing in an appendix or supplementary material.*

*In general, I found it challenging to follow the entire pre-processing workflow. I think the authors should work on improving this section's readability, making each step taken very clear so that other users can replicate the workflow."*

**Response:**

Thank you for your comment. We think it is important to keep these steps in the Main Text to enable a full understanding of the steps taken, and the logical order needs to be maintained to ensure that it can be followed. The pre-processing is fundamentally different to the post-processing (section 4.3), for example, even though some of the same data sets are used in each. However, we have taken your point on board and attempted to rewrite this section in order to make it more readable.

**Comment #8**

*"Page 10, Line 213*

*The last line of the paragraph can be removed."*

**Response:**

Thank you. We have updated the manuscript.

**Comment #9**

*"Page 21*

*Most of the links included in Table A1 are not working."*

**Response:**

Thank you for checking. This is unfortunately the nature of non-persistent URLs. We have included all of the data in the Zenodo repository. It is out of our control to ensure the URLs are persistent for individual sources and so we list the URLs we accessed, even if currently no longer active.